# Planting time shapes fall armyworm infestation dynamics and associated yield loss of maize in Bangladesh

Md Mostafizur Rahman Shah[ID][1]*, Most. Sirajum Munira[1], Md Forhad Hossen[1], Alison Watson[2]

1 IPM Laboratory, Division of Entomology, Bangladesh Wheat and Maize Research Institute, Dinajpur, Bangladesh, 2 AgriFood Systems CoLab, ASEAN FAW Action Plan, Adjunct Fellow, Macquarie University

* mostafiz.wrc@gmail.com

## Abstract

In Bangladesh, maize is usually cultivated from early November to mid-December, however, recent intensification of cropping systems has expanded its cultivation period from October to March to accommodate region-specific patterns. The invasion of fall armyworm (FAW), *Spodoptera frugiperda* (J.E. Smith), in 2018 has since posed a serious threat to maize cultivation. This study assessed the seasonal dynamics of FAW infestation and its effects on maize under IPM and control conditions across six planting months over three consecutive years in the Old Himalayan Piedmont Plain (AEZ 1), Bangladesh. Field experiments followed a split-plot design with sowing month as the main plot and treatment as the subplot. Two treatments were applied: (i) IPM plots received seed treatment with cyantraniliprole followed by foliar applications of biopesticides, and insecticides; and (ii) an untreated control. Leaf infestation was assessed from V4-V12 stages, and cob damage, grain yield, and yield loss were recorded at harvest. Heatmap analysis showed consistently high infestation in control plots, peaking at 99.3% during February-March, whereas IPM plots maintained low infestation (<4%). Infestation in untreated control plots exhibited a U-shaped, non-linear response to planting time, described by a cubic polynomial model ($R^2 = 0.456$), with highest levels in October and March and lowest in December. Cob damage and damage intensity mirrored infestation patterns, with maximum damage in March (20.3%) and significantly higher values in control plots than IPM ones. Grain yield was significantly influenced by planting month and treatment, and their interaction. Yield loss also varied across planting month, with the highest losses in March (31.6%) and February (19.4%) and minimal losses during November-January (<3.3%). These findings demonstrate that combining October planting with the recommended IPM approach maximizes maize production, whereas planting from October to January without intervention helps reduce fall armyworm (FAW) infestation and maintain yield stability in Bangladesh.

**Data availability statement:** All relevant data are within the paper and its Supporting Information files.

**Funding:** The Ministry of Agriculture provides yearly budget to the Bangladesh Wheat and Maize Research Institute (BWMRI) like other research institutions in Bangladesh. There is no specific budget separately for this study but from the total budget of BWMRI we got the financial and material support for the study which is incorporated in the acknowledgement section. It is noted that the funders had no role in study design, data collection and analysis, decision to publish, or preparation of the manuscript, and did not receive salary by any authors.

**Competing interests:** The authors have declared that no competing interests exist relating to the research, authorship, or publication of this manuscript.

## Introduction

Maize (*Zea mays* L.), a member of the family Poaceae, is one of the most important cereal crops globally. It is cultivated in over 170 countries, covering an estimated 197 million hectares, with a global production of 1163.5 million tons [1]. The global utilization pattern of maize indicates that approximately 56% is used as animal feed, 13% for food, and 31% for industrial purposes [2]. In developing countries, lower- and middle-income countries account for nearly two-thirds of maize production, hence maize plays a vital role in food security and the livelihoods of smallholder farmers. Given the projected global population of 9.3 billion by 2050, demand for maize in developing countries is expected to more than double [3].

In Bangladesh, maize is considered as one of the most important cereal crops after rice and wheat due to its wide adaptability, high productivity, and growing demand [4,5]. Beyond its growing use as feed in poultry, dairy, and aquaculture industries, maize consumption in the form of processed products such as popcorn, sweet corn, baby corn, and corn-based snacks is also increasing and production has expanded rapidly [6]: maize cultivation area increased from 0.33 million ha in 2017−18 to nearly 0.63 million ha in 2023−24, while production rose from 2.89 to 6.70 million tons [7]. This consistent growth underscores the rising significance of maize in Bangladesh's agricultural landscape.

Despite its expansion, maize production faces multiple challenges. Studies in South Asia have consistently identified input shortages (seed, fertilizer, irrigation) and pest infestations as major constraints [8,9]. In Bangladesh, insect pests remain a persistent problem, with surveys by the Department of Agricultural Extension (DAE) ranking pest attack as the most significant barrier to maize cultivation, reported by over 86% of field officials [10]. Key insect pests include cutworm, armyworm, corn earworm, and stalk borer. Other pests identified include white grubs, corn seed maggot, corn leaf aphid and corn wireworm as well as white grubs, grasshopper, termite, chinch bug, seed corn beetle, corn rootworm, stink bug and thrips. More recently, the fall armyworm (FAW), *Spodoptera frugiperda* (J.E. Smith, 1797) (Lepidoptera: Noctuidae) represents a major threat, assisted by favourable environmental and climatic conditions in Bangladesh.

The fall armyworm is a highly polyphagous pest, native to the Americas, with records of more than 350 host plants across 76 families, predominantly Poaceae, Asteraceae, and Fabaceae [11]. Its preferred hosts include maize, rice, sorghum, sugarcane, and cotton. FAW is newly invasive in Bangladesh, having spread rapidly after being first reported in India and Bangladesh in 2018 [12,13]. The pest has since become established across South and Southeast Asia, where it has caused significant economic losses, particularly in maize [14].

FAW caterpillars are highly voracious and can attack maize at all developmental stages, which can lead to substantial yield losses. Reported yield reductions vary by infestation level and crop stage [15,16]. Studies in Africa and Asia further highlight yield losses of 30–40%, with localized severe infestations reducing biomass, leaf area, and grain yield by up to 40% [17–19]. Farmers frequently rely on chemical

insecticides for management, but this has led to concerns around resistance development and negative environmental and health impacts [20–23].

Farmers understandably often panic when seeing substantial foliar damage from FAW infestations, however, such damage may not always result in large yield losses as maize can recover well from early leaf feeding [15]. It is the crop stage that is more important than the level of visible injury: early vegetative damage usually has less effect, while infestations during tasseling and grain filling will significantly reduce grain yield [16,24]. Therefore, relying only on leaf damage scores tends to overestimate economic loss. Management decisions should thus account for both crop stage and pest density.

In Bangladesh, although FAW is now recognized as a major pest, quantitative assessments of its impact on maize yield remain scarce. Given the pest's rapid spread and the conducive agro-climatic conditions in the country, systematic studies are required to determine the extent of yield losses and explore management options. In particular, cultural practices such as adjusting planting time may play a critical role in reducing FAW damage by disrupting its life cycle and reducing crop exposure during peak infestation periods [53]. The objectives of this study are to: (i) quantify the seasonal dynamics of FAW infestation across six planting months; (ii) evaluate the efficacy of an integrated pest management (IPM) strategy; and (iii) determine the associated yield losses under field conditions in northwestern Bangladesh.

## Materials and methods

The study was conducted at the Bangladesh Wheat and Maize Research Institute (BWMRI) research field located in Dinajpur district (Latitude 25.74° and Longitude 88.67°), with the permission of the Director General of BWMRI, which did not involve endangered or protected species. Dinajpur is the largest maize cultivating district in Bangladesh among 64 districts over last five years [25]. Study periods were 2020−21, 2021−22, and 2022−23 years where maize was sown on the 15th (±2 days) of October, November, December, January, February, and March of each year. The experimental site is situated in Agroecological Zone 1 (AEZ 1), also known as the Old Himalayan Piedmont Plain in northwestern Bangladesh. The climate is characterized by moderate to high rainfall during the monsoon season (July-August), a mild winter from November to February with cooler temperatures prevailing in December-January that favor Rabi (winter) crop cultivation, and a hot, humid summer from March to June. The weather conditions during the study period are presented in Fig 1.

### Experimental design and treatments applied

The experimental plots were designed as split plots where the maize sowing month was maintained as the main plot and the FAW management treatment was placed in the subplot with 3 replications (S1 Fig). The unit plot size was $8m \times 6.6m$ with a maintained spacing of 20 cm seed-to-seed and 60 cm row-to-row. In the main plot, plants were sown in six different planting months, i.e., October, November, December, January, February, and March. The treatment factor had two levels $T_1$ = a full IPM schedule (seed treatment with cyantraniliprole combined with foliar applications of *Spodoptera frugiperda* Nuclear Polyhedrosis Virus (SfNPV) based biopesticides, emamectin benzoate and spinosad) and $T_2$ = an untreated control. In the IPM plots, the SfNPV-based biopesticide (Fawligen; $7.5 \times 10^9$ virus inclusion bodies mL$^{-1}$, applied at 0.4 mL L$^{-1}$ of water) was applied twice as foliar sprays at 7-day intervals. Fawligen applications were initiated once FAW larvae at 1st-2nd instar and infestation level reached at 10−20%. This period typically occurred 20−22 days after sowing in October, 30−35 days after sowing from November to January, and 18−20 days after sowing during February-March. Upon monitoring, when infestation level again approached >20% and the larvae at 3rd-4th instar, generally at the V6-V8 growth stages, emamectin benzoate (proclaim 5SG applied at 1 gm L$^{-1}$ of water) was applied once and followed spinosad (success 2.5SC @1.3 ml L$^{-1}$ of water) once at 7 days interval. These insecticides were selected based on prior efficacy studies [26] and are classified as moderately to slightly hazardous (WHO Class II–III) [27]. In total, four foliar sprays were performed during study period.

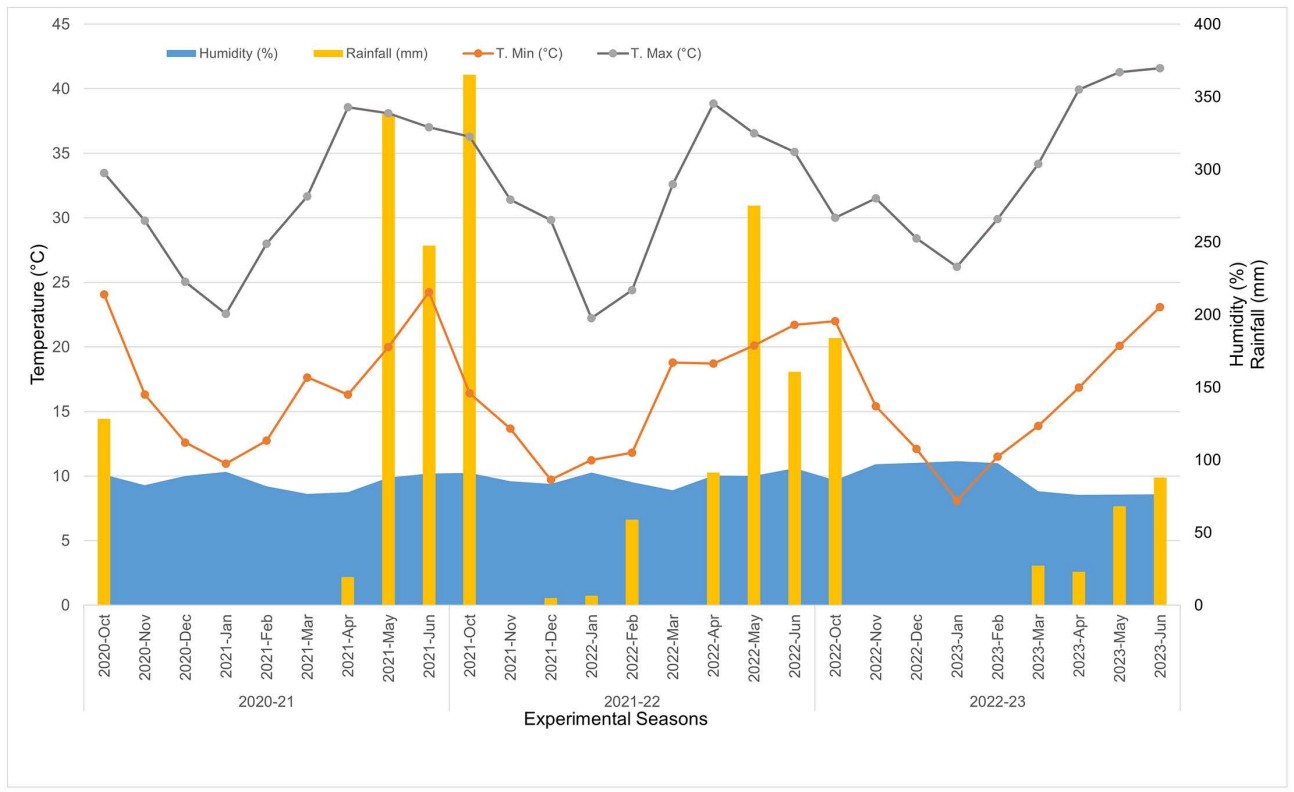

**Fig 1. Monthly variation in minimum and maximum temperature (°C), rainfall (mm), and relative humidity (%) during the maize growing seasons from 2020–21 to 2022–23 at the experimental site.**

## Crop management

The experiment was conducted using the hybrid maize variety Pioneer 3355 across three consecutive growing seasons, 2020−21, 2021−22, and 2022−23. Fertilization was applied according to the recommended nutrient rates set by BWMRI, ensuring optimal plant growth. The specific nutrient application included nitrogen (N) at 250 kg ha$^{-1}$, phosphorus (P) at 60 kg ha$^{-1}$, potassium (K) at 110 kg ha$^{-1}$, sulfur (S) at 47 kg ha$^{-1}$, magnesium (Mg) at 12 kg ha$^{-1}$, zinc (Zn) at 3.6 kg ha$^{-1}$, and boron (B) at 1.7 kg ha$^{-1}$. Throughout the crop growth period, irrigation was supplied as needed to maintain adequate soil moisture, and weeding was conducted regularly to minimize competition from unwanted plants and ensure optimal maize development.

## Leaf and Cob infestation assessment

FAW infestation data were obtained on-site after the initial infestation (10–20%) was observed. Data was collected using the "W" pattern approach, with five spots selected in each plot and ten plants assessed in each spot (S2 Fig). This is important to note that the fall armyworm, *Spodoptera frugiperda*, collected from the study field was identified through genomic DNA (gDNA) analysis of the mitochondrial COI region. Later, periodic collection and morphological identification continued throughout the study period, during which typical larvae showed a distinct inverted "Y" head mark and four large dark spots arranged in a square on the upper side of the last but one abdominal segment. Further, host plant feeding pattern with irregular "windowing" and ragged holes found in leaves or whorl, heavy feeding in young whorls with fresh, moist, and abundant granular frasses (pale to dark brown) dumped near feeding sites or in the whorl/ear confirm the presence of

FAW. In the cobs, feeding at the tip first, frass-filled cavities among kernels near the tip; husk pierced with small holes, and frass often visible (Fig 2). Leaf infestation data were recorded from the V4 to V12 stages by examining the upper three leaves, where new infestations were consistently observed on freshly emerged foliage. The percentage of cob damage and the intensity of cob damage (1–9 scale) were measured [28].

### Yield and yield attributes assessment

After the cob had matured, samples were gathered from 2m, 3 rows (2m x 1.8m = 3.6 m2) of maize plants in the field. Three sampling points were selected from each plot following a diagonal transect, extending from one corner to the opposite corner of the plot, to ensure representative coverage of the entire experimental area. The mean number of plants per unit area (3.6m$^2$) was determined for each experimental plot to assess plant density and distribution. Cobs were obtained from plants taken in a 3.6m$^2$ unit area. After shucking, harvested cobs were counted. The grains were deshelled after two to three days of drying. The grains were weighed after being sun-dried again until just 14% moisture remained. The yield values were converted to kilogram per hectare after being determined for a 3.6m$^2$ unit area.

### Statistical analysis

The variables were analyzed using mixed-model ANOVA in JMP statistical software 13[th] version [54], with planting date as the main-plot factor and treatment as the sub-plot factor, while year and replication were considered random effects in the model. Model residuals were examined to verify ANOVA assumptions. Normality was assessed using the Shapiro-Wilk test [55] and visual inspection of normal quantile (Q-Q) plots, while homogeneity of variances was evaluated using Levene's test [56]. Assumptions were considered satisfied when residuals showed no substantial deviations from normality or

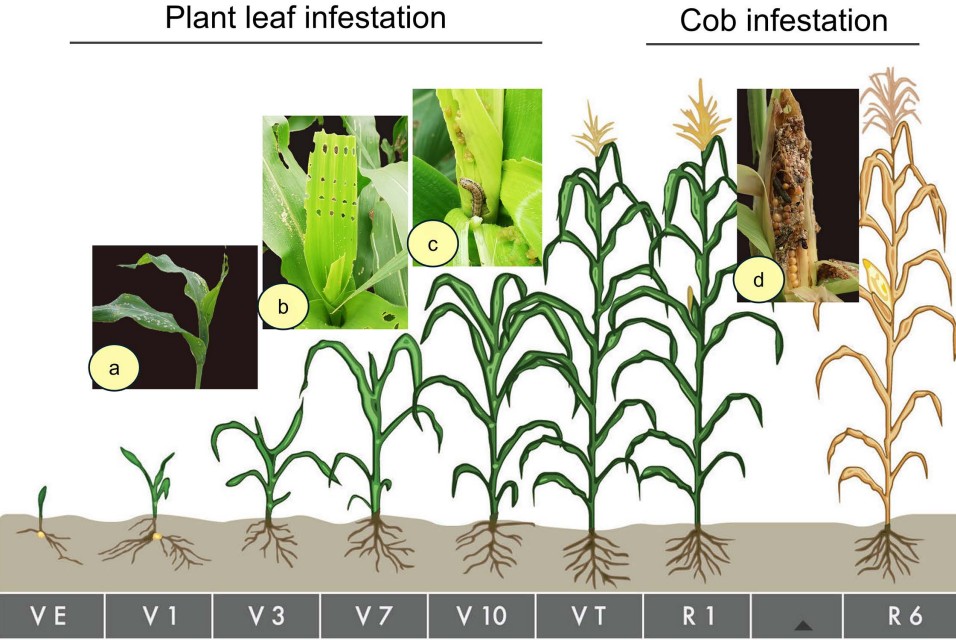

**Fig 2. Arbitrary phenological growth stages of maize showing the leaf and cob infestation damage symptoms caused by *S. frugiperda* in Bangladesh.** Feeding pattern: (a) Small Fresh Windowing (SFW) by 1-2 instar larvae (b) Ragged linear holes by 3-4 instar larvae (c) Infested Whorl (IW) caused by mostly 5-6 instar larvae, and **(d)** Enormous cob damage by 1-6 instar larvae.

variance homogeneity. The effects of individual factors, as well as their interactions, were evaluated. Mean values were separated using Tukey's HSD at $P = 0.05$ when significant variable effects were detected, while comparisons between the IPM and control were performed using Student's t-test. All figures, except the heatmap, were generated using JMP software, whereas the heatmap was constructed in R (version 4.4.2).

## Results

The incidence of fall armyworm (FAW) in maize varied significantly by month, treatment, and their interaction.

### Leaf damage

The heatmap shows the infestation level of fall armyworm (FAW), which is significantly varied between treatments across planting months corresponding to various vegetative stages of the maize plant (V4: $F_{5,524} = 14.569$, $P < 0.0001$; V6: $F_{5,524} = 48.648$, $P < 0.0001$; V8: $F_{5,524} = 33.607$, $P < 0.0001$; V10: $F_{5,524} = 51.797$, $P < 0.0001$; V12: $F_{5,524} = 98.499$, $P < 0.0001$) (Fig 3). The heatmap visually represents the percentage of plants infested, with a color gradient where darker purple indicates low/zero infestation and yellow/orange indicates high infestation (Fig 3).

Under control conditions, infestation was relatively moderate in October (42–68% at V4–V12), declined markedly during November–January (generally ≤20%), and then increased sharply in the month of February and March. The highest infestation remained consistent in March, particularly at later vegetative stages (V6–V12), where infestation recorded 62–78%. February also showed elevated infestation, especially at V8-V12 (63–86%). In contrast, the IPM treatment consistently maintained lower infestation levels across all months and stages. During November–January, infestation remained minimal (mostly <7%). Even during peak infestation months (February–March), IPM plots limited

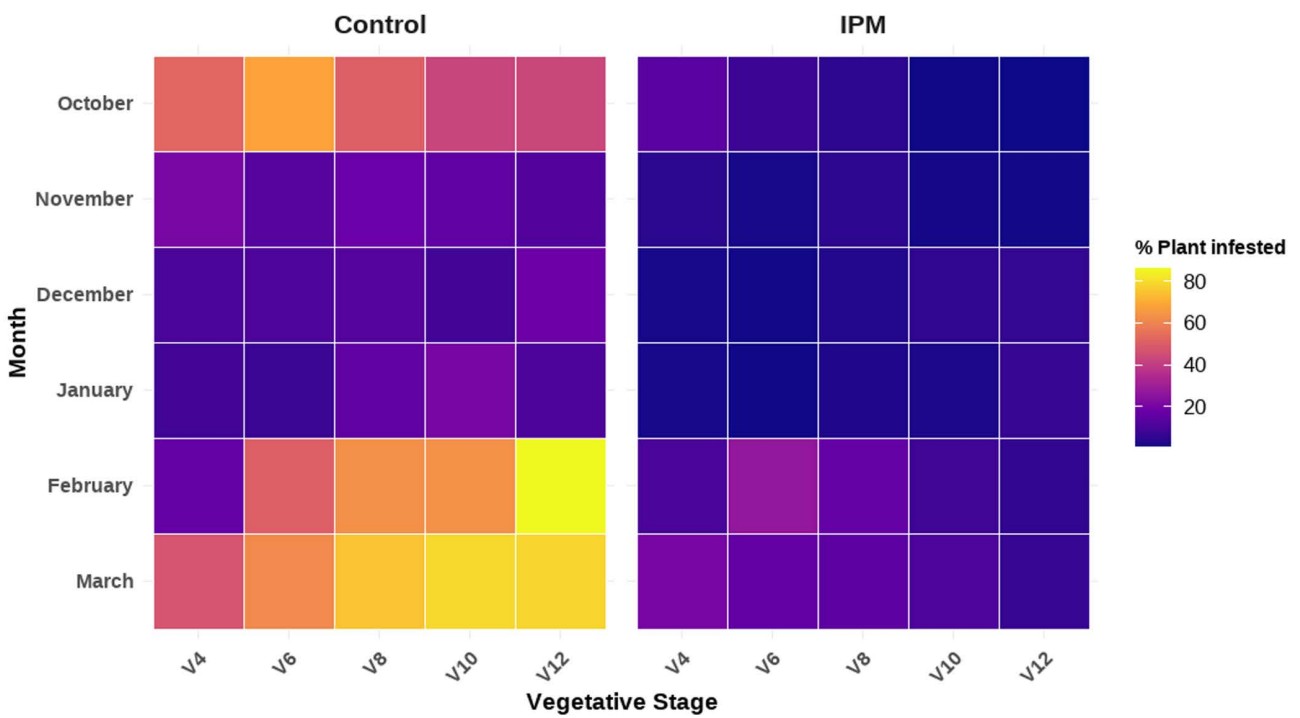

**Fig 3. Heat map illustrating the percentage of maize plants infested by *S. frugiperda* at different vegetative stages (V4–V12) across planting months under control and IPM treatments.**

infestation to approximately 6–27%, substantially lower than the corresponding control plots. In most months, infestation tended to increase with crop growth stage, reaching its peak at V8–V12 and infestation remained suppressed under IPM plot (Fig 3; S1 & S2 Tables).

**Fall armyworm infestation dynamics over time in untreated control**

The relationship between planting month and fall armyworm infestation in control plots is described by a cubic polynomial regression model (Fig 4). In this study, the cubic equation of best fit is $Y = 52.21 - 56.97 \times Month + 21.21 \times Month^2 - 1.817 \times Month^3$. Where $Y$ is the predicted % infestation, and month is the numerical representation of the planting month. The model demonstrated a moderate fit to the data, with a coefficient of determination ($R^2$) of 0.456. This indicates that approximately 45.6% of the variation in fall armyworm infestation can be explained by the variation in the planting month. The cubic fit line clearly illustrates a significant, U-shaped, non-linear trend in fall armyworm infestation levels across the observed planting months (October to March). The observed U-shaped pattern might be partially driven by other unexplained factors, including abiotic conditions (e.g., humidity and rainfall) and biotic interactions (e.g., natural enemies and other insects). However, infestation levels were high for early planting, starting around 68% in October. The infestation rate dropped sharply, reaching its minimum level (7–8%) in the plots planted in December. This indicates that planting maize in December was associated with the lowest incidence of fall armyworm. Following the minimum in December, the infestation rate showed a sharp and continuous increase. Infestation climbed through January and February, peaking at the highest observed levels (approaching 75%) for the crops planted in March (Fig 4).

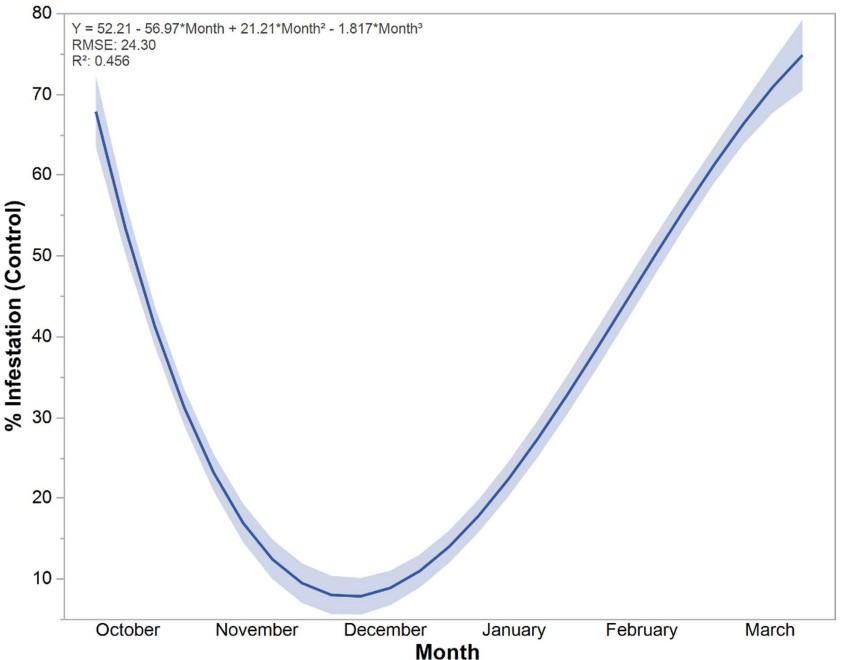

**Fig 4. Cubic line of fit demonstrates plant leaf infestation dynamics by *S. frugiperda* across planting month in control condition in Bangladesh.**

## Effect of treatments

The mean percentage of infested plants across six planting months, based on all growth stages, is shown in violin and box plots (Fig 5). Results showed that the percentage of infested plants varied significantly between IPM and control treatments ($F_{1, 2684} = 1673.511$, $P < 0.0001$). The untreated control had a mean infestation of 36.1%, whereas the IPM plot exhibited a markedly lower infestation rate of 7.16% (red dashed lines). This represents an approximate 80% reduction in leaf infestation under IPM relative to the control. The distribution of infestation levels further highlighted clear differences between IPM and control plots. In control plots, infestation ranged from 0 to 100%, with a median near 30%, whereas in IPM plots, the distribution is heavily skewed and concentrated below 10%, with the median closest to zero. A wide inter-quartile range (IQR) was observed in the control, spanning 10–60%, whereas the IPM showed an IQR of 0–10% (box plot, Fig 5). High infestation intensities (≥60%) were frequently observed in the control treatment; in contrast, IPM recorded minimal damage, although rarely higher infestation values (up to ~80%) were observed (black dots, Fig 5).

## Cob damage

Cob damage was significantly influenced by treatment and planting month. Treatment had a highly significant effect on cob damage ($F_{1,308} = 118.996$, $P < 0.0001$), demonstrating that the IPM package substantially reduced FAW-induced cob injury compared to the untreated control. Seasonal variation in cob damage differed markedly among the six planting months both in control ($F_{5,152} = 18.327$; $P < 0.0001$) and IPM ($F_{5,152} = 2.482$; $P = 0.034$) plots (Fig 6). In the control plots, cob damage varied significantly across the six planting months ($F_{5,152} = 18.33$, $P < 0.0001$), indicating a strong seasonal influence on infestation intensity. The highest cob damage was observed in March (33.6%), which was statistically similar to February (26.9%), whereas the lowest cob damage was observed in the remaining months, October, November, December, and January (ranged 11.5–15.4%) (Fig 6). In the IPM plot, although seasonal differences were also statistically significant, the magnitude of variation was considerably smaller ($F_{5, 152} = 2.48$, $P = 0.034$). Cob damage remained relatively low throughout the season, ranging from 2.7 to 10.6%. Unlike the control, no sharp late-season surge was observed under IPM management.

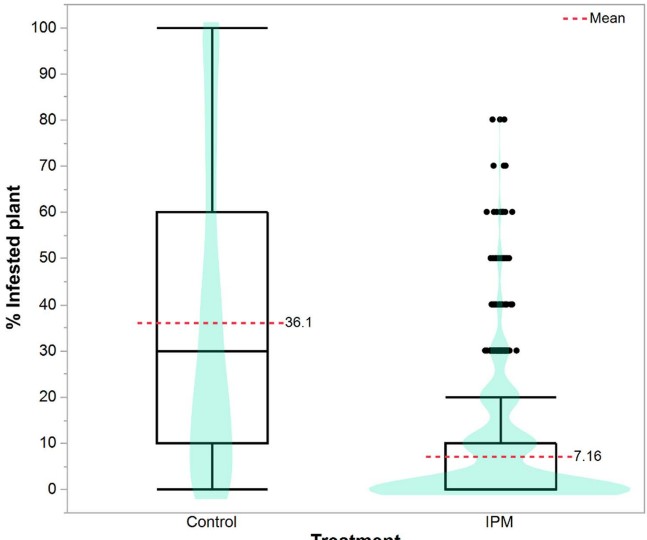

**Fig 5. The violin and box plot represents the comparison of *S. frugiperda* leaf infestation levels in maize between control and IPM treatments.**

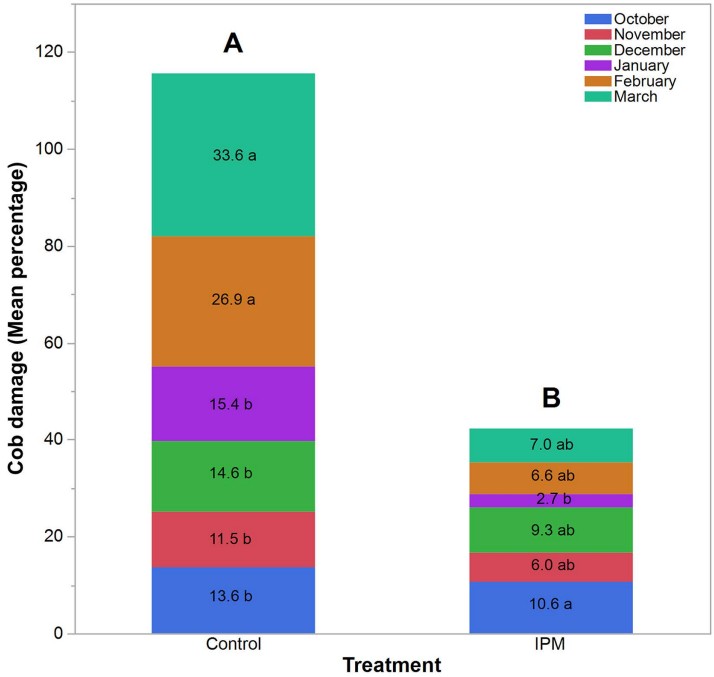

**Fig 6. Stacked bar graph showing the percentage of cob damage in maize infested by *S. frugiperda* across planting months under control and IPM treatments.** Comprising the same letter are not significantly differed according to Tukey's HSD at α = 0.05.

### Cob damage intensity (19 scale)

The intensity of cob damage closely followed overall cob damage trends, showing significant variation by treatment and planting month. Treatment effects on cob damage intensity were highly significant ($F_{1,308} = 188.350$, $P < 0.0001$), with the IPM plots exhibiting substantially lower damage intensity than the control plots (Fig 7). Monthly variation in the untreated control was highly significant ($F_{5,152} = 52.188$, $P < 0.0001$), with values of 3.0 (October), 2.0 (November), 2.3 (December), 2.7 (January), 5.52 (February), and peaking at 6.63 in March on the 1–9 scale (Fig 7). In contrast, though seasonal differences were statistically significant in the IPM treatment ($F_{5,152} = 6.00$, $P < 0.0001$), the magnitude of variation was comparatively small, and no sharp late-season surge was observed (1.6–2.6 on the same scale).

### Plant number and cob number

Analysis of variance showed significant effects of month on both plant number ($F_{5,308} = 5.495$, $P < 0.0001$) and cob number ($F_{5,308} = 12.925$, $P < 0.0001$), while treatment (IPM vs control) did not have a significant effect on neither plant number ($F_{1,308} = 1.153$, $P = 0.2838$) nor cob number ($F_{1,308} = 0.728$, $P = 0.3939$). The month × treatment interaction was significant for plant number ($F_{5,308} = 2.595$, $P = 0.0256$) but not for cob number ($F_{5,308} = 0.797$, $P = 0.5524$) (Table 1; S3 Table).

By month, the highest number of plants was recorded in January (27.1±0.2 plants per 3.6m2), which was statistically similar to December and October. The lowest plant number was observed in March (25.5±0.4) and February (25.6±0.3), though these are statistically similar to that of October and November. Cob numbers peaked in November (30.2±0.6) but were not statistically different from those in October and January. The lowest cob counts were recorded in February and March, while December and January showed intermediate (Table 1; S3 Table).

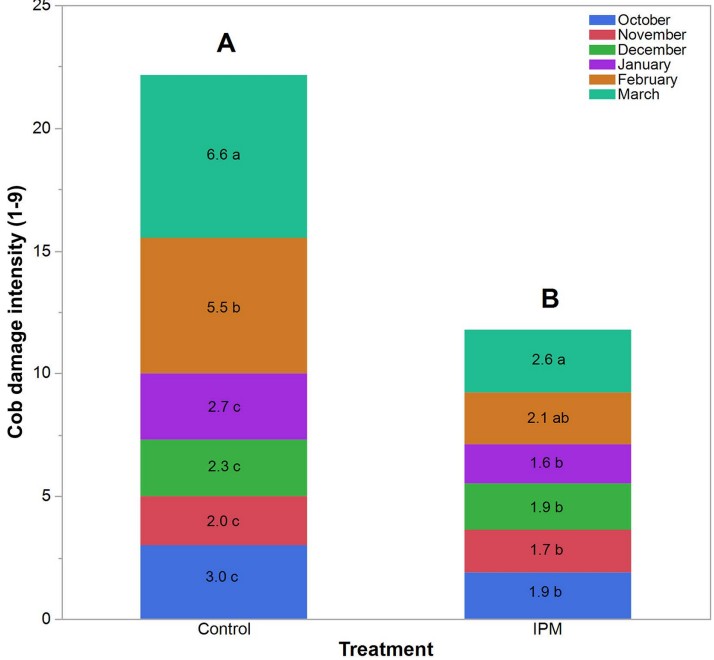

**Fig 7. Stacked bar graph illustrates the cob damage intensity (1-9 scale) in maize infested by *S. frugiperda* across planting months under control and IPM treatments.** Comprising the same letter are not significantly differed according to Tukey's HSD at α = 0.05.

**Table 1. Effect of planting months and treatments on plant number and cob number (mean ± SE) of maize per 3.6m² infested by *S. frugiperda*.**

| Variables | Plant no. per 3.6m² | Cob no. per 3.6 m² |
|---|---|---|
| **Month** | | |
| October | 26.5 ± 0.4 abc | 29.7 ± 1.0 ab |
| November | 25.7 ± 0.3 bc | 30.2 ± 0.6 a |
| December | 26.9 ± 0.2 ab | 27.5 ± 0.6 bc |
| January | 27.1 ± 0.2 a | 28.0 ± 0.3 ab |
| February | 25.6 ± 0.3 c | 25.6 ± 0.3 c |
| March | 25.5 ± 0.4 c | 25.5 ± 0.4 c |
| **Treatment** | | |
| IPM | 26.3 ± 0.2 | 27.9 ± 0.4 |
| Control | 26.1 ± 0.2 | 27.6 ± 0.4 |

Values in columns are not separated by sources of variation and comprising the same letter are not significantly differed according to Tukey's HSD at α = 0.05.

## Maize yield and yield loss due to FAW infestation

Maize yield in both IPM and control plots varied significantly across planting months, and yield loss due to fall armyworm (FAW) infestation also differed markedly among months (Table 2). Planting month had a highly significant effect on grain yield in both IPM ($F_{5,152} = 52.036$, $P < 0.0001$) and control ($F_{5,152} = 114.963$, $P < 0.0001$) plots. In the IPM plots, the highest yield was recorded in October (14,134.3 ± 637.7 kg ha⁻¹), which was not statistically different from November and

**Table 2. Effect of planting months on maize yield under IPM and control treatments, and yield loss caused by *S. frugiperda*.**

| Variables | Yield (kg ha⁻¹) | | Yield loss (%) |
|---|---|---|---|
| | IPM | | Control |
| **Month** | | | |
| October | 14134.3±637.7 a | 12734.9±577.9 a | 9.9±0.6 c |
| November | 13244.4±285.8 ab | 12967.8±313.1 a | 2.2±0.6 d |
| December | 13889.0±241.9 ab | 13437.7±247.6 a | 3.3±0.6 d |
| January | 12821.7±104.8 b | 12397.2±88.5 a | 3.2±0.8 d |
| February | 10744.6±161.0 c | 8655.3±135.1 b | 19.4±0.8 b |
| March | 8556.4±239.1 d | 5917.3±249.7 c | 31.6±1.4 a |
| **Treatment** | | | |
| IPM | 12239.2±203.1 a | | 11.6 |
| Control | 11018.4±251.8 b | | |

Values in columns are not separated by sources of variation and comprising the same letter are not significantly differed according to Tukey's HSD at α=0.05.

December. Yield declined gradually in later plantings, with the lowest yield observed in March (8,556.4±239.1 kg ha⁻¹) (Table 2).

A similar trend was observed in the control plots, where the highest yields were obtained from December (13,437.7±247.6 kg ha⁻¹), which is statistically similar to that of October, November, and January, and while the lowest yield occurred in March (5,917.3±249.7 kg ha⁻¹). In the treatment effect, the IPM plot produced significantly ($F_{1,308}$=53.089, $P<0.0001$) higher yield (12,239.2±203.1 kg ha⁻¹) compared to the control (11,018.4±251.8 kg ha⁻¹). The interaction between planting month and treatment was also significant ($F_{5,308}$=5.989, $P<0.0001$), indicating that the effect of IPM on yield differed across planting months (S3 Table). Yield loss due to FAW infestation was also significantly affected by planting month ($F_{5,44}$=120.483, $P<0.0001$). The lowest yield losses were recorded in November (2.23±0.64%), which is statistically similar to December and January, while moderate losses occurred in October (9.95±0.57%). In contrast, substantially higher yield losses were observed in late plantings, particularly in February (19.38±0.80%) and March (31.57±1.41%), indicating increased FAW pressure in later planting periods (Table 2).

## Discussion

The findings of this study provide critical insights into the impact of planting time on the invasion and resulting yield loss of maize due to the fall armyworm, *S. frugiperda* (JE Smith), in the context of Bangladesh. The results clearly demonstrate that planting month is a dominant factor influencing FAW infestation, severity of damage, and ultimately, maize yield loss. The temporal dynamics of FAW incidence, as revealed by heatmap and regression analyses, showed that infestation intensity was strongly dependent on seasonal variation.

Heatmap showed that the pronounced peaks observed in February and March coincide with favorable climatic conditions, including higher temperatures with later vegetative stages, increased maize canopy that provide shelter and food resources for larvae. In contrast, the low incidence recorded during December and January corresponds to cooler and drier conditions, which are likely to constrain FAW survival and reproduction.

The cubic regression model establishing a U-shaped trend between planting month and FAW infestation is the most striking finding regarding pest dynamics. Infestation was highest for the earliest (October) and latest (February, March) planting dates, warmer months than November to January. On the other hand, infestation reaching a minimum during December (7–8% infestation), November to January are cooler months in Bangladesh. Previous research has

demonstrated that FAW infestation and its population dynamics are strongly associated with weather factors, with temperature emerging as one of the most critical determinants [29,30].

The dramatically lower infestation in December-planted crops (Fig 4) provides less/no management recommendation for smallholder farmers. Crops planted in the mid-winter months (November-January) likely benefit from lower ambient temperatures, not suitable for FAW population build-up, can slow the reproductive cycle, as well as lead to phenological asynchrony with major immigration events. Conversely, the sharp surge in infestation from February onwards, peaking in March, coincides with rising temperatures and the crop's entry into vulnerable later vegetative stages (V8-V12), creating a 'perfect storm' for severe damage. These findings are consistent with previous observations that FAW activity increases with temperature and vegetative stages, ensuring food availability [24,31–33].

The comparison between IPM schedule and untreated control treatments clearly demonstrated the effectiveness of the applied IPM in suppressing FAW infestation. IPM plots maintained consistently low infestation levels (<5%), with only minor peaks under severe pest pressure. In contrast, untreated control plots exhibited severe infestations exceeding 90% during high-risk months (February-March). These findings confirm that the IPM scheduled, seed treatment with cyantraniliprole, twice foliar application of *Spodoptera frugiperda* Nuclear Polyhedrosis Virus (SfNPV), and once each of emamectin benzoate and spinosad provided substantial protection against FAW infestation. To promote insecticide resistance management (IRM) and ensure long-term sustainability, the insecticides applied in this study were rotated among different mode-of-action groups as part of a season-long integrated pest management (IPM) program. However, occasional infestations in IPM plots indicate that even effective management measures may be challenged under extreme pest pressure or prolonged infestation windows with some limitations likely foliar spray in proper time. Similar results have been documented for various management approaches, including biopesticide use, seed treatment with cyantraniliprole, and IPM strategies [26,34–37]. Further, the economic analysis incorporated the costs of all insecticides and labor required for IPM implementation, which were approximately USD 100 ha$^{-1}$ higher than those of the control treatment. Under the Bangladesh market context, this additional cost corresponds to the value of approximately 375 kg of maize grain. Therefore, when the yield loss exceeds 375 kg ha$^{-1}$, the adoption of control measures against fall armyworm can be considered economically viable, as the monetary value of the yield saved outweighs the additional management costs. In the present study, IPM application was economically profitable for maize planted in October, February, and March. In contrast, December and January plantings were only marginally profitable, whereas November planting resulted in a negative economic return from the treatment.

The pattern of cob damage mirrors the leaf infestation dynamics. Monthly variation was observed where late-sown maize, February and March, was more prone to higher levels of cob damage. These findings align with reports from Africa and Asia indicating that FAW abundance is strongly associated with temperature and host availability, with peak infestations typically occurring during warmer months when late-season maize provides abundant green foliage [38,39]. The increased damage intensity in late-sown crops may also reflect overlapping generations of the pest, as continuous planting sustains population buildup across the growing season. In contrast, early sown crops, such as those planted in November, December or January likely escape the pest pressure due to asynchronous development with FAW population.

Treatment effects were highly significant, demonstrating the efficacy of the applied pest management approach in reducing both the incidence and severity of FAW-induced cob damage. The IPM plots recorded >60% reduction in cob damage compared to the control plot, confirming the practical value of the intervention. Similar reductions in FAW infestation through the use of seed treatment, biological control agents, or selective insecticides have been documented in previous studies [26,37,40,41]. These findings reinforce the importance of IPM strategies that combine preventive measures with timely intervention to suppress FAW populations.

Interestingly, while the IPM plots experienced markedly lower cob damage, seasonal trends in infestation intensity remained evident, suggesting that environmental conditions and planting time still play critical roles in modulating pest pressure. Therefore, integrating control measures with optimized planting schedules could provide a synergistic benefit.

Early sowing, coupled with the use of recommended IPM schedule, may help maize crops escape the most damaging FAW generations, thereby enhancing yield stability.

The analysis of plant and cob numbers showed that the treatment itself did not significantly affect overall plant or cob establishment as a main effect. This is important as it indicates the applied treatment combination is not phytotoxic. Whereas sowing month significantly influenced maize plant establishment and cob number. The strong influence of temporal factors suggests that variations in environmental conditions, particularly temperature, rainfall, and soil moisture at planting, played a major role in determining plant stand establishment and cob formation. Similar findings have been reported by [42,43], who observed that early-season weather variability can markedly affect maize germination, growth uniformity, and reproductive success. Seasonal rainfall distribution and optimal temperature during crop establishment may have contributed to improved plant stands and higher cob formation. Previous studies across South Asia have emphasized that maize performance is highly sensitive to the timing and distribution of rainfall, as moisture stress during vegetative growth or flowering can reduce cob initiation and kernel set [44,45].

Maize yield and yield loss due to *S. frugiperda* infestation were significantly influenced by planting month. Overall, the early sowing months October, November, and December recorded significantly higher yields with comparatively lower yield losses, while late sowings, particularly in February and March, exhibited drastic reductions in yield. This could be attributed to the fact that maize sown between October and December experiences a longer vegetative stage under relatively low pest pressure, resulting in higher yields. In contrast, crops planted in February and March undergo a shorter vegetative phase and suffer higher levels of infestation on both leaves and cobs, leading to reduced yields. Research evidence indicates that altering sowing dates can substantially influence maize yield outcomes [46–48]. It is noteworthy that under Bangladesh climatic conditions, maize can be harvested by the end of May (±10 days), regardless of whether planting occurs in October or March. March sowing resulted in the highest yield loss (31.6%), followed by February (19.4%), reflecting the severe impact of FAW infestation during late planting warmer periods. Other studies have shown that heavy FAW pressure during late-season plantings reduces kernel filling and grain weight, resulting in significant yield losses [49–52]. "Our results align with recent findings from East Africa, where Cokola et al. [53] reported severe FAW infestations in late-planted maize. Our study builds on this by quantifying the consequent yield loss gradient and demonstrating that an IPM regimen can mitigate, though not eliminate, the risk associated with suboptimal planting times.

The consistently higher yields under IPM plots compared to control ones across all sowing months indicate the efficacy of pest management interventions in minimizing FAW damage. However, even in IPM plots, yield reductions during late sowings highlight that chemical or biological control alone may not fully offset the adverse effects of late planting.

It is important to note that this study evaluated a bundled IPM package rather than the individual components in isolation. The individual contribution of each intervention- seed treatment, biopesticide or synthetic insecticides to the overall efficacy against *S. frugiperda* cannot be disaggregated from our results. The bundled design limits our ability to determine the relative cost-effectiveness for smallholder farmers where resource constraints are critical. Another limitation of this study is that the results were obtained from a single research station using only one maize hybrid. Future research should therefore employ component-wise or factorial experimental designs to deconstruct the IPM protocol and disentangle the individual and interactive effects of each management strategy across multiple locations.

## Conclusion

The study conclusively demonstrates that planting month is a dominant factor influencing fall armyworm (*S. frugiperda*) infestation and subsequent maize yield loss in Bangladesh. The U-shaped temporal dynamic of FAW infestation, with peaks in the warmest months (October, February, March) and a critical minimum during mid-winter (December-January), provides a powerful, actionable insight for smallholder farmers. Crops planted in the cooler months of November to January significantly benefit from conditions unfavorable for FAW population buildup, leading to substantially lower leaf and cob damage and higher yields. The integrated pest management (IPM) strategy, including seed treatment with cyantraniliprole

and foliar applications of SfNPV based biopesticides at earlier stages and less hazardous insecticides, emamectin benzoate and spinosad at later stages, proved highly effective, consistently keeping infestation levels low in IPM compared to control plots. The synergistic combination of October planting with the recommended IPM approach represents the most robust strategy for maximizing maize production. In contrast, adjusting the planting window from October to January, in the absence of intervention, appears to be a suitable strategy for reducing fall armyworm (FAW) infestation and maintaining stable maize yields. Adopting such integrated, seasonally adaptive strategies will be crucial for sustainable maize production and resilience against FAW invasion in South Asia.

## Supporting information

**S1 Fig. Experimental layout showing the split-plot design, with maize sowing month assigned to the main plots and *S. frugiperda* management treatments allocated to the subplots, replicated across three blocks.** (TIF)

**S2 Fig. Schematic representation of leaf infestation data collection using the "W" sampling pattern, with five sampling points selected per plot and ten maize plants assessed at each point.** (TIF)

**S1 Table. Mean (± SE) leaf infestation (%) on maize by *Spodoptera frugiperda* as influenced by the Month × Treatment interaction, with means grouped using Tukey's HSD test.** (DOCX)

**S2 Table. ANOVA results (mean ± SE) of leaf infestation across planting months and treatments of different vegetative growth stages in maize under *Spodoptera frugiperda* infestation, with means grouped using Tukey's HSD test.** (DOCX)

**S3 Table. Mean (± SE) plant number, cob number, and yield of maize as influenced by the Month × Treatment interaction under *Spodoptera frugiperda* infestation, with means grouped using Tukey's HSD test.** (DOCX)

**S1 File. Striking Image. Arbitrary phenological growth stages of maize showing the leaf and cob infestation damage symptoms caused by S. *frugiperda* in Bangladesh.** Feeding pattern: (a) Small Fresh Windowing (SFW) by 1–2 instar larvae (b) Ragged linear holes by 3–4 instar larvae (c) Infested Whorl (IW) caused by mostly 5–6 instar larvae, and (d) Enormous cob damage by 1–6 instar larvae. (TIF)

## Acknowledgments

We extend our sincere appreciation to all staff members of the IPM Laboratory, Entomology Division, BWMRI, for their valuable assistance and dedicated support throughout the study. We gratefully acknowledge the Ministry of Agriculture, Government of the People's Republic of Bangladesh, for providing the research facilities necessary to conduct the study.

## Author contributions

**Conceptualization:** Md Mostafizur Rahman Shah.

**Data curation:** Md Mostafizur Rahman Shah, Most. Sirajum Munira, Md Forhad Hossen.

**Formal analysis:** Md Mostafizur Rahman Shah, Most. Sirajum Munira, Md Forhad Hossen.

**Funding acquisition:** Md Mostafizur Rahman Shah.

**Investigation:** Md Mostafizur Rahman Shah.

**Methodology:** Md Mostafizur Rahman Shah.

**Project administration:** Md Mostafizur Rahman Shah.

**Resources:** Md Mostafizur Rahman Shah.

**Software:** Md Mostafizur Rahman Shah, Md Forhad Hossen.

**Supervision:** Md Mostafizur Rahman Shah.

**Validation:** Md Mostafizur Rahman Shah.

**Visualization:** Md Mostafizur Rahman Shah.

**Writing – original draft:** Md Mostafizur Rahman Shah.

**Writing – review & editing:** Md Mostafizur Rahman Shah, Most. Sirajum Munira, Md Forhad Hossen, Alison Watson.

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
