## [Decision Letter · Decision Letter 0]

2 Feb 2026

Dear Dr. Shah,

Thank you for submitting your manuscript to PLOS ONE. After careful consideration, we feel that it has merit but does not fully meet PLOS ONE’s publication criteria as it currently stands. Therefore, we invite you to submit a revised version of the manuscript that addresses the points raised during the review process.

Please address all comments made by the three reviewers (note that most of Reviewer 1 comments are in the download file). If you disagree with the reviewer please provide a detailed defense. I have included in "additional editor comments" below my concerns that I consider most important.

We look forward to receiving your revised manuscript.

Kind regards,

Rodney N. Nagoshi, Ph.D.

Academic Editor

PLOS One

Journal Requirements:

“We gratefully acknowledge the Ministry of Agriculture, Government of the People’s Republic of Bangladesh, for providing financial support for this research.”

“We gratefully acknowledge the Ministry of Agriculture, Government of the People’s Republic of Bangladesh, for providing financial support for this research.”

Additional Editor Comments:

I have four primary areas of concern and also have one observation that may deserve a comment.

Major concerns:

1. This paper claims to be specific to FAW damage. However many other pests of maize are identified in Bangladesh (lines 70-73) and it is not stated in the Methods how damage specific to FAW was identified (noted by Reviewers 1 and 3). How did you distinguish plants/cobs damaged by FAW from those affected by other pests?

2. All three reviewers (particularly Reviewer 3) suggested modifications of the statistical analyses. Please address their concerns.

3. As noted by Reviewer 3, much is made about the impact of treatment (lines 406-418). However, from an economic perspective the most relevant metric is yield. Table 2 shows consistent yield loss between treated and untreated, but I see no analysis testing statistical significance. Please provide that comparison. The relevant question is whether any improvement in yield produced by treatment is likely to be greater than the cost of treatment. I don't expect a detailed economic analysis but it would at minimum be useful to know whether treatment significantly improves yield.

4. The conclusion (lines 534-546) recommends planting in Nov-Jan combined with treatment. Yet in Table 2, planting in October with treatment gave the highest yield. Doesn't this suggest that planting in October with treatment is the most productive strategy? Seems to me that your recommendation only makes sense for non-treatment plots where the highest yields occur in Nov-Dec. Please comment.

Observation: I noted in Table 2 that highest yield occurred in October in treated plots but shifted to December in untreated plots. Why this shift? Do you believe it is of significance?

Reviewers' comments:

Reviewer's Responses to Questions

**Comments to the Author**

1. Is the manuscript technically sound, and do the data support the conclusions?

Reviewer #1: Yes

Reviewer #2: Yes

Reviewer #3: Partly

2. Has the statistical analysis been performed appropriately and rigorously?

Reviewer #1: Yes

Reviewer #2: Yes

Reviewer #3: No

3. Have the authors made all data underlying the findings in their manuscript fully available?

Reviewer #1: Yes

Reviewer #2: Yes

Reviewer #3: Yes

4. Is the manuscript presented in an intelligible fashion and written in standard English?

Reviewer #1: Yes

Reviewer #2: Yes

Reviewer #3: Yes

Reviewer #1: Comments are on the attached file. Since fall armyworm is an invasive pest, the authors should describe how its feeding and damage are different from traditional maize pests in Bangladesh. The cost effectiveness of the seed treatment and foliar sprays should be mentioned and if maize harvested at the times in the study have a market.

Reviewer #2: General comments

This manuscript investigates how planting time influences Fall Armyworm (Spodoptera frugiperda) infestation and associated yield loss in maize in Bangladesh. The study is well structured and methodologically sound, addressing an important agronomic challenge in South Asia. The use of a three-year split-plot design with treated and untreated plots yields robust data, and the findings are presented with clear statistical analysis. Identifying a U-shaped infestation pattern and quantifying yield loss across planting months are valuable contributions to integrated pest management (IPM) strategies in the region.

However, the manuscript would benefit from being placed in a broader context within the existing literature. The impact of an otherwise strong study is slightly limited by the absence of key comparisons and some methodological justifications. Specific comments for improvement are provided below.

Abstract

Ensure the research gap is clearer. Add a sentence about why planting time studies are needed in Bangladesh.

Lines 23-25: The description of the treatment is dense. Consider simplifying to: "... treated plots received seed treatment with Cyantraniliprole followed by foliar applications of biopesticides and insecticides; and (ii) an untreated control."

Line 29: The term "innate leaf infestation" is confusing. Specify that it refers to "infestation in untreated control plots."

Lines 38-39: The conclusion is vague. Strengthen it with a practical recommendation: "These findings demonstrate that adjusting planting time to November-January, combined with the described IPM strategy, is an effective approach to minimize FAW-related yield loss in Bangladesh."

Introduction

Lines 95–102: While the rationale for studying planting time is clear, the objectives are not explicitly stated. Add a final sentence: 'The objectives of this study are to: (i) quantify the seasonal dynamics of FAW infestation across six planting months; (ii) evaluate the efficacy of an integrated pest management (IPM) strategy; and (iii) determine the associated yield losses under field conditions in northwestern Bangladesh.'

Lines 100-102 (Gap in Literature): This section omits a highly relevant study by Cokola et al. (2024), published in PLoS ONE, which examined the effects of planting date on FAW in the Democratic Republic of Congo. Referring to this study is crucial for contextualising the global significance and originality of your findings in Bangladesh.

Materials and methods

Lines 129-143 (Treatment application protocol): The protocol is complex. A supplementary flowchart (S1 Fig.) would greatly improve clarity, showing the decision points (e.g., "IF infestation reaches 10-20%, apply SfNPV...").

Lines 140-142: Justify the selection of these specific insecticides (Emamectin benzoate, Spinosad) briefly. E.g., "based on prior efficacy studies and their classification as moderately hazardous (WHO Class II)."

Lines 190-199 (Statistical Analysis): The analysis is appropriate. However, specify how data normality and homogeneity of variances were checked prior to ANOVA. All statistical analyses and figures were performed using JMP statistical software, version 13 but we can see that some figures were made using R. Also, mention the specific R version used for the heatmap.

Missing Statement: An ethics/field permission statement should be added at the end of the manuscript: "Field studies were conducted with the permission of the Bangladesh Wheat and Maize Research Institute (BWMRI) and did not involve endangered or protected species."

Results

The results are clearly presented and statistically robust, with effective visualizations (e.g. heatmaps and regression models) that demonstrate the significant differences in FAW infestation and yield outcomes between management regimes.

However, there is a significant inconsistency between the methodology description and the results presentation regarding treatment structure, which requires clarification and potentially affects the interpretation of the findings.

In the methods section (lines 129–143): The treatment is described as a sequential, integrated protocol: Seed treatment with cyantraniliprole, followed by foliar sfNPV (biopesticide), emamectin benzoate and spinosad (applied based on infestation thresholds). In the results section: Data are presented only in two aggregated categories: 'Treated' vs. 'Non-treated'.

This raises three critical questions:

1. Attribution of efficacy: Which component(s) of the multi-step protocol were most responsible for the observed suppression of FAW? Was it the seed treatment, the biopesticide, the synthetic insecticides, or a combination of these? The current presentation cannot answer this.

2. Scientific and practical value: Presenting only the combined 'Treated' result limits the study's value. A key research question, for example, is whether the biopesticide (SfNPV) could provide sufficient early-season control alone or in initial combination with the seed treatment, potentially reducing the need for synthetic insecticide sprays later on. The current design and analysis cannot address this.

3. Statistical design vs. analysis: The manuscript describes a split-plot design with 'treatment' as a subplot factor. If the 'treatment' factor had only two levels (treated/non-treated), the analysis would be correct. However, the methodology suggests that multiple distinct interventions were applied sequentially. If these were applied to the same plots, they would not be independent treatments, but rather parts of a schedule. This should be made explicit to avoid confusion.

Recommendations for revision:

In the methods: Clarify the experimental design. State explicitly: 'The treatment factor had two levels: (1) a full IPM schedule (seed treatment and sequential foliar sprays, as described) and (2) an untreated control. If any other treatment combinations were tested (e.g. seed treatment only or biopesticide only), these must be reported.

In the discussion: Acknowledge this limitation. Add a statement such as: 'It is important to note that our study evaluated a bundled IPM package. The individual contribution of the seed treatment, biopesticide or synthetic insecticide to the overall efficacy cannot be disaggregated from our results. Future research should deconstruct this protocol to identify the most cost-effective components for smallholder farmers.'

For future work: Suggest that follow-up studies include factorial designs that test the components separately (e.g. seed treatment or biopesticide alone or in combination) to determine their individual and interactive effects.

Figure and table citations are present (e.g. Fig. 1, Table 1), but the captions are either incomplete or embedded in the text.

Figure 5: The percentage of maize plants infested per treatment is missing. It should be two combined figures.

On lines 246–248, when reporting the cubic model (R² = 0.456), it would be accurate to note that this indicates considerable unexplained variance. It would be accurate to briefly mention that other factors (e.g. humidity, natural enemies) also play a role.

Figure 7 caption: Do not combine damage intensity and % cob damage in the same figure. Instead, use a stacked bar plot with the treatments on the x-axis and the years in the legend. This will result in four combined figures.

Discussion

Lines 322–333 (Contextualization): The discussion must reference and discuss the work of Cokola et al. (2024). Add a paragraph: “Our results align with recent findings from East Africa, where Cokola et al. (2024) reported severe FAW infestations in late-planted maize. Our study builds on this by quantifying the consequent yield loss gradient and demonstrating that an IPM regimen can mitigate, though not eliminate, the risk associated with suboptimal planting times.”

Missing section: Limitations. A dedicated paragraph is required to address the following:

Single-site, single-variety limitation: The findings are from one research station with one hybrid.

Economic feasibility: The cost-effectiveness of the recommended IPM package for smallholders was not analysed.

Mechanistic inference: Temperature is cited as the driver, but local FAW life-table parameters were not measured.

Lines 413–415 (Insecticide Use): Promote Insecticide Resistance Management (IRM). Add: 'To ensure sustainability, the insecticides used here should be rotated with insecticides from different mode-of-action groups as part of a season-long IPM programme.'

Lines 529–546 (Conclusion): The claim that planting month is the 'single most dominant factor' (line 529) is strong, but it could be tempered to 'a dominant factor' unless it is statistically compared to all other variables.

Reviewer #3: • In text, fall armyworm does not require capitalization.

• Generic chemical names – cyantraniliprole, emamectin benzoate, and spinosad – do not require capitalization in text.

• Line 70: and stalk borer.

• Line 134: what was the larval stage at each application? Fawligen is ineffective past the 4th instar.

• Lines 157 – 187: The manuscript attributes foliar and ear injury to FAW, but no information is provided on whether larvae were sampled to confirm species identity. Because other lepidopteran pests can produce similar feeding symptoms in maize, especially at the whorl and ear stages, some confirmation of FAW presence (even periodic field collections) would strengthen the species-specific conclusions. If such sampling was conducted, it should be described. If not, the authors should clarify that damage assessments were based on characteristic FAW injury and temper species-level inferences accordingly.

• Line 189-199: The study addresses an important question and the data appear potentially valuable. However, the current statistical analysis does not fully account for the split-plot design and multiple years of study. Because planting date is the main effect and treatment is the subplot effect, a mixed-model ANOVA would be more appropriate. In that case, planting date would be the main effect, treatment subplot effect, and year and replicate included as random effects. Additionally, subsamples within plots would not have to be averaged prior to analysis as the model can account for within-plot variability using replicate as a random effect.

• The results of the treatment effect are being overstated in the discussion. The primary finding is that planting date significantly affected yield, yield loss, plant, and cob numbers. In tables 1 and 2, the only time that treatment was a significant effect was in the month x treatment interaction. And in table 2, the only months with significant differences in yield were October, November, and December, with Nov and Dec having lower infestations than any other time of year. This discussion may change with the mixed-model ANOVA, but I would caution the authors to not over-emphasize the significance of treatment based on this data.

**Do you want your identity to be public for this peer review?** For information about this choice, including consent withdrawal, please see our For information about this choice, including consent withdrawal, please see our Privacy Policy .

Reviewer #1: **Yes:** Robert L MeagherRobert L Meagher

Reviewer #2: **Yes:** Marcellin Cuma CokolaMarcellin Cuma Cokola

Reviewer #3: No

---

## [Author Response · Author response to Decision Letter 1]

16 Mar 2026

Date: 16 March 2026

To

Rodney N. Nagoshi, Ph.D.

Academic Editor

PLOS One

Ref: PONE-D-26-00624

Title: Planting time shapes Fall Armyworm infestation dynamics and associated yield loss of maize in Bangladesh

Dear Editor

Thank you so much for sending the comments raised by the academic editor and reviewer(s) which have allowed us to considerable improvements to the article. We are happy to inform you that we have been able to resolve almost all issues raised by the academic editor and both reviewers. If unfortunately, we have overlooked any aspects kindly let us know and we will rectify. Kindly note that all edits are available in the text in track change mode. Response for all comments raised by all reviewers and the handling editor are available below:

Journal Requirements:

Authors’ response: We meet the PLOS ONE’s style requirements including file naming.

Authors’ response: Thanks for your comment and we mentioned the full name of authority in the materials and methods section.

“We gratefully acknowledge the Ministry of Agriculture, Government of the People’s Republic of Bangladesh, for providing financial support for this research.”

Authors’ response: We have removed funding statements from the acknowledgement section and amended statements included in the cover letter.

“We gratefully acknowledge the Ministry of Agriculture, Government of the People’s Republic of Bangladesh, for providing financial support for this research.”

Authors’ response: Actually, the Ministry of Agriculture provides yearly budget to the Bangladesh Wheat and Maize Research Institute (BWMRI) like other research institutions in Bangladesh. There is no specific budget separately for this study but from the total budget of BWMRI we got the financial and material support for the study which is incorporated in the acknowledgement section.

Authors’ response: We comply with the publisher’s open data policy and, during resubmission, we will revise the data availability statement to ensure that the study data are freely accessible.

Authors’ response: Thank you for this guidance. We carefully reviewed the reviewer-recommended publications to assess their relevance to our study. We confirm that all citations added in the revised version were selected based on their scientific relevance and contribution to the topic.

Additional Editor Comments:

I have four primary areas of concern and also have one observation that may deserve a comment.

Major concerns:

1. This paper claims to be specific to FAW damage. However many other pests of maize are identified in Bangladesh (lines 70-73) and it is not stated in the Methods how damage specific to FAW was identified (noted by Reviewers 1 and 3). How did you distinguish plants/cobs damaged by FAW from those affected by other pests?

Authors’ response: Thank you for your valid question. In the revised version, clearly describe FAW specific damage symptoms in the methods section (leaf and cob infestation assessment subsection).

2. All three reviewers (particularly Reviewer 3) suggested modifications of the statistical analyses. Please address their concerns.

Authors’ response: The suggested modifications have been addressed and edited the revised version.

3. As noted by Reviewer 3, much is made about the impact of treatment (lines 406-418). However, from an economic perspective the most relevant metric is yield. Table 2 shows consistent yield loss between treated and untreated, but I see no analysis testing statistical significance. Please provide that comparison. The relevant question is whether any improvement in yield produced by treatment is likely to be greater than the cost of treatment. I don't expect a detailed economic analysis but it would at minimum be useful to know whether treatment significantly improves yield.

Authors’ response: Thank you for this guidance. Suggested issue has been incorporated into the revised version.

4. The conclusion (lines 534-546) recommends planting in Nov-Jan combined with treatment. Yet in Table 2, planting in October with treatment gave the highest yield. Doesn't this suggest that planting in October with treatment is the most productive strategy? Seems to me that your recommendation only makes sense for non-treatment plots where the highest yields occur in Nov-Dec. Please comment.

Authors’ response: Thank you for this direction and suggested concern has been incorporated.

Observation: I noted in Table 2 that highest yield occurred in October in treated plots but shifted to December in untreated plots. Why this shift? Do you believe it is of significance?

Authors’ response: Higher yields were obtained when IPM was combined with October planting, likely because crops planted in October experienced a longer vegetative growth period. In contrast, under the absence of FAW management, December planting produced higher yields, which corresponded with the relatively lower fall armyworm infestation observed during that period. It is noted that although the maize planting times differed, the harvesting times were nearly similar, with only a 10–20 day difference (last of May to early of June) between the earliest and the latest plantings.

Reviewers' comments:

Reviewer's Responses to Questions

Comments to the Author

1. Is the manuscript technically sound, and do the data support the conclusions?

Reviewer #1: Yes

Reviewer #2: Yes

Reviewer #3: Partly

2. Has the statistical analysis been performed appropriately and rigorously?

Reviewer #1: Yes

Reviewer #2: Yes

Reviewer #3: No

3. Have the authors made all data underlying the findings in their manuscript fully available?

Reviewer #1: Yes

Reviewer #2: Yes

Reviewer #3: Yes

4. Is the manuscript presented in an intelligible fashion and written in standard English?

Reviewer #1: Yes

Reviewer #2: Yes

Reviewer #3: Yes

5. Review Comments to the Author

Reviewer #1: Comments are on the attached file. Since fall armyworm is an invasive pest, the authors should describe how its feeding and damage are different from traditional maize pests in Bangladesh. The cost effectiveness of the seed treatment and foliar sprays should be mentioned and if maize harvested at the times in the study have a market.

Authors’ response: All concern comments provided in separate sheet have been responded to the revised manuscript.

Reviewer #2: General comments

This manuscript investigates how planting time influences Fall Armyworm (Spodoptera frugiperda) infestation and associated yield loss in maize in Bangladesh. The study is well structured and methodologically sound, addressing an important agronomic challenge in South Asia. The use of a three-year split-plot design with treated and untreated plots yields robust data, and the findings are presented with clear statistical analysis. Identifying a U-shaped infestation pattern and quantifying yield loss across planting months are valuable contributions to integrated pest management (IPM) strategies in the region.

Authors’ response: Thank you for your valuable comments, which we greatly appreciate and which encourage us to improve the manuscript.

However, the manuscript would benefit from being placed in a broader context within the existing literature. The impact of an otherwise strong study is slightly limited by the absence of key comparisons and some methodological justifications. Specific comments for improvement are provided below.

Abstract

Ensure the research gap is clearer. Add a sentence about why planting time studies are needed in Bangladesh.

Authors’ response: The suggested edits have been incorporated.

Lines 23-25: The description of the treatment is dense. Consider simplifying to: "... treated plots received seed treatment with Cyantraniliprole followed by foliar applications of biopesticides and insecticides; and (ii) an untreated control."

Authors’ response: The suggested edits have been incorporated.

Line 29: The term "innate leaf infestation" is confusing. Specify that it refers to "infestation in untreated control plots."

Authors’ response: The suggested edits have been incorporated.

Lines 38-39: The conclusion is vague. Strengthen it with a practical recommendation: "These findings demonstrate that adjusting planting time to November-January, combined with the described IPM strategy, is an effective approach to minimize FAW-related yield loss in Bangladesh."

Authors’ response: The suggested edits have been incorporated into the revised manuscript.

Introduction

Lines 95–102: While the rationale for studying planting time is clear, the objectives are not explicitly stated. Add a final sentence: 'The objectives of this study are to: (i) quantify the seasonal dynamics of FAW infestation across six planting months; (ii) evaluate the efficacy of an integrated pest management (IPM) strategy; and (iii) determine the associated yield losses under field conditions in northwestern Bangladesh.'

Authors’ response: The suggested edits have been incorporated into the revised manuscript.

Lines 100-102 (Gap in Literature): This section omits a highly relevant study by Cokola et al. (2024), published in PLoS ONE, which examined the effects of planting date on FAW in the Democratic Republic of Congo. Referring to this study is crucial for contextualising the global significance and originality of your findings in Bangladesh.

Authors’ response: Thank you for your suggestion. The suggested edits have been incorporated into the revised manuscript.

Materials and methods

Lines 129-143 (Treatment application protocol): The protocol is complex. A supplementary flowchart (S1 Fig.) would greatly improve clarity, showing the decision points (e.g., "IF infestation reaches 10-20%, apply SfNPV...").

Authors’ response: The suggested edits have been incorporated into the revised manuscript.

Lines 140-142: Justify the selection of these specific insecticides (Emamectin benzoate, Spinosad) briefly. E.g., "based on prior efficacy studies and their classification as moderately hazardous (WHO Class II)."

Authors’ response: The suggested edits have been incorporated into the revised manuscript.

Lines 190-199 (Statistical Analysis): The analysis is appropriate. However, specify how data normality and homogeneity of variances were checked prior to ANOVA. All statistical analyses and figures were performed using JMP statistical software, version 13 but we can see that some figures were made using R. Also, mention the specific R version used for the heatmap.

Authors’ response: The suggested edits have been incorporated into the revised manuscript.

Missing Statement: An ethics/field permission statement should be added at the end of the manuscript: "Field studies were conducted with the permission of the Bangladesh Wheat and Maize Research Institute (BWMRI) and did not involve endangered or protected species."

Authors’ response: The suggested edits have been incorporated into methods section of the re

---

## [Editor Report · Decision Letter 1]

30 Mar 2026

Planting time shapes fall armyworm infestation dynamics and associated yield loss of maize in Bangladesh

PONE-D-26-00624R1

Dear Dr. Shah,

We’re pleased to inform you that your manuscript has been judged scientifically suitable for publication and will be formally accepted for publication once it meets all outstanding technical requirements.

Kind regards,

Rodney N. Nagoshi, Ph.D.

Academic Editor

PLOS One
---

## [Editor Report · Acceptance letter]

PONE-D-26-00624R1

PLOS One

Dear Dr. Shah,

I'm pleased to inform you that your manuscript has been deemed suitable for publication in PLOS One. Congratulations! Your manuscript is now being handed over to our production team.

Kind regards,

on behalf of

Dr. Rodney N. Nagoshi

Academic Editor

PLOS One